# Mouse Breast Carcinoma Monocytic/Macrophagic Myeloid-Derived Suppressor Cell Infiltration as a Consequence of Endothelial Dysfunction in *Shb*-Deficient Endothelial Cells Increases Tumor Lung Metastasis

**DOI:** 10.3390/ijms222111478

**Published:** 2021-10-25

**Authors:** Qi He, Maria Jamalpour, Eric Bergquist, Robin L. Anderson, Karin Gustafsson, Michael Welsh

**Affiliations:** 1Department of Medical Cell Biology, Uppsala University, Box 571, Husargatan 3, 75123 Uppsala, Sweden; heqinju@gmail.com (Q.H.); mariajamalpour1@gmail.com (M.J.); Eric.Bergquist.5099@student.uu.se (E.B.); 2Translational Breast Cancer Program, Olivia Newton-John Cancer Research Institute, Heidelberg 3084, Australia; robin.anderson@onjcri.org.au; 3School of Cancer Medicine, La Trobe University, Bundoora 3083, Australia; 4Harvard Stem Cell Institute, Cambridge, MA 02138, USA; UGUSTAFSSON@mgh.harvard.edu; 5Massachusetts General Hospital, 185 Cambridge Street, Boston, MA 02114, USA

**Keywords:** breast carcinoma, metastasis, angiogenesis, immune suppression, SHB

## Abstract

Metastasis reflects both the inherent properties of tumor cells and the response of the stroma to the presence of the tumor. Vascular barrier properties, either due to endothelial cell (EC) or pericyte function, play an important role in metastasis in addition to the contribution of the immune system. The *Shb* gene encodes the Src homology-2 domain protein B that operates downstream of tyrosine kinases in both vascular and immune cells. We have investigated E0771.lmb breast carcinoma metastasis in mice with conditional deletion of the *Shb* gene using the *Cdh5-Cre^ERt2^* transgene, resulting in inactivation of the *Shb*-gene in EC and some hematopoietic cell populations. Lung metastasis from orthotopic tumors, tumor vascular and immune cell characteristics, and immune cell gene expression profiles were determined. We found no increase in vascular leakage that could explain the observed increase in metastasis upon the loss of *Shb* expression. Instead, *Shb* deficiency in EC promoted the recruitment of monocytic/macrophagic myeloid-derived suppressor cells (mMDSC), an immune cell type that confers a suppressive immune response, thus enhancing lung metastasis. An MDSC-promoting cytokine/chemokine profile was simultaneously observed in tumors grown in mice with EC-specific *Shb* deficiency, providing an explanation for the expanded mMDSC population. The results demonstrate an intricate interplay between tumor EC and immune cells that pivots between pro-tumoral and anti-tumoral properties, depending on relevant genetic and/or environmental factors operating in the microenvironment.

## 1. Introduction

Metastasis is the most common cause of death in patients diagnosed with breast cancer [1]. Metastasis may occur following either hematogenous or lymphatic dissemination. Hematogenous metastasis requires tumor cell intravasation by cells crossing the blood barrier [2]. Once in the blood stream, the tumor cells must seed in distant organs of which the lung is a common site for breast cancer metastasis [1]. Tumor cell seeding also requires tumor cells to cross the vascular barrier at the site of metastasis [2]. Consequently, the integrity of the vascular barrier is important in the regulation of metastasis [3]. The integrity of endothelial cell (EC) junctions and pericyte coverage of small vessels are both important in restricting leakage and consequently metastasis [3,4]. Vascular endothelial growth factor-A (VEGFA) exerts a major influence on junction leakiness [5,6,7] and, given its high level of production in tumors, there will be an increased propensity towards tumor vascular leakiness, facilitating the metastatic process.

Additionally, the endothelium regulates immune cell infiltration [8]. This is particularly relevant for tumor growth and metastasis since hematopoietic cells (HC) constitute major components of the tumor microenvironment. In a rapidly expanding tumor, the macrophage compartment is typically pro-tumoral while anti-tumoral T cell responses would be expected to be suppressed [9]. Tumor T cell suppression commonly results from activation of CD4^+^/Forkhead box P3 (FoxP3^+^) T regulatory cells (Tregs) that inhibit effector T cells by various mechanisms [10,11,12]. Myeloid-derived suppressor cell (MDSC) are also immunosuppressive through various means, including inhibition of interferon-gamma (IFNγ) expression, increased interleukin-10 (IL-10) production, and stimulation of Tregs, all contributing to an immune-suppressed local environment [13]. This will largely result from the local cytokine/chemokine milieu.

The Src homology-2 domain protein B (SHB) is an adapter protein operating downstream of several tyrosine kinase receptor that exerts pleiotropic effects [14]. The *Shb* gene is required for VEGFA-induced angiogenesis and vascular leakage [15,16,17,18], pericyte coverage [18,19] and immune responses [19,20,21,22]. In tumor biology, *Shb* exerts multiple effects, depending on tumor type. RIP-Tag insulinomas, T241 fibrosarcomas and Lewis lung carcinomas exhibit reduced growth and vascular density when grown in mice lacking *Shb* [15,16]. B16F10 melanomas show increased metastasis in the absence of *Shb* [18,19]. Conditional deletion of *Shb* in EC or pericytes exerted different effects on tumor growth, metastasis and the vasculature [18]. Decreased pericyte coverage, increased leakage and metastasis were observed in mice with *Shb*-deficient pericytes [18] whereas reduced tumor growth and vascular leakage were detected in EC *Shb* deficient mice [18], indicating different roles of SHB in these vascular cell types. Finally, in 4T1 breast carcinomas, *Shb* deficiency appears to primarily exert its effects via immune cell responses [21]. However, since those experiments were performed using a global *Shb* knockout mouse, indirect effects via other cell types could not be excluded.

Since the *Shb-*gene exerts regulatory roles in both EC and HC in a tumor context, and since breast cancer growth and metastasis are influenced by both of these stromal cell types, we considered employing the EC-specific *Shb-*gene knockout model to obtain a better understanding of the role of these cell types and their interplay in promoting/suppressing tumor expansion. For this purpose, we assessed E0771.lmb lung metastasis in mice with endothelial *Shb* deficiency. We found that the immune system plays a major role in suppressing metastasis and that *Shb-*deficient EC promote the recruitment or expansion of an immune-suppressed tumor microenvironment.

## 2. Results

### 2.1. Tumor Growth and Metastatic Lung Colonization

E0771.lmb tumor cells were orthotopically injected into wild type mice (*Cdh5-Cre^ERt2^*) or mice with the *Shb* gene conditionally deleted in EC (*Shb^flox/flox^/Cdh5-Cre^ERt2^*). This Cre transgene also deletes *Shb* in some HC [22]. There was no difference in tumor growth until Day 14 (Figure 1), at which ulcerations started to appear over some tumors, prompting tumor resection. When the mice were sacrificed three weeks later, visual inspection of lungs revealed increased lung surface metastasis in mice with *Shb* absent in EC (Figure 1).

### 2.2. Tumor Vasculature

The tumor vasculature was investigated since vascular leakage is important for metastasis, and the *Shb* gene has been shown to influence several EC and pericyte parameters. In agreement with previous studies, the absence of *Shb* in EC reduced vascular density and leakage (Figure 2). No effect on pericyte coverage of small vessels was noted (Figure 2). Consequently, the data provide no support for an impaired vascular barrier as an explanation for the observed increase in metastasis.

### 2.3. Tumor Immune Responses

Considering the possibility that tumor immune responses may play a role in tumor metastasis, we set out to characterize the immune cell profiles of tumor CD45^+^ cells from wild type (WT) and EC *Shb* KO mice using flow cytometry. Gating strategies are shown in Appendix A. We noted an increased CD11b^+^/Ly6C^+^/CD14^+^ population (mMDSC or monocytic/macrophagic myeloid-derived suppressor cells), both as a percentage of parental population (CD11b^+^) and in total cell numbers, in KO tumor CD45^+^ cells (Figure 3). This was accompanied by increased percentages of CD4^+^ and Treg (FoxP3^+^ of CD4^+^) cells (Figure 3). Local (inguinal) lymph nodes showed a concomitant decrease in their Treg population (Figure 3), suggesting an increased transfer of this cell population to the tumor. Other immune cell populations determined (CD11b^+^, CD8^+^ of CD11b^+^, CD11c^+^, CD8^+^ of CD11c^+^, CD11b^+^/Ly6G^+^ double positive, and CD11b^+^/Ly6G^+^/CD15^+^ = nMDSC or polymorphonuclear myeloid-derived suppressor cells) were not altered (Table 1).

To determine if these changes would reflect the situation at the lung metastatic seeding sites, we performed immune profiling of lung CD45^+^ cells in parts of the lungs heavily colonized by tail vein injected tumor cells. As was the case in the primary tumor, the areas colonized by tumor cells exhibited an increased percentage of mMDSC cells (Figure 4) without the presence of Tregs being affected.

### 2.4. Tumor CD45^+^ Gene Expression

To understand the increased recruitment of mMDSC in tumors grown in mice with *Shb* conditionally deleted in EC, we isolated tumor-derived CD45^+^ cells, performed qPCR for a number of potentially relevant genes, and compared the results with those of HC-deleted *Shb* using *Vav1-Cre* as a control to exclude off-target effects of *Cdh5-Cre^ERt2^* in HC. Thus, gene changes specific to *Cdh5-Cre^ERt2^* would reflect only cell-autonomous effects of *Shb* in EC. For certain genes (*Il10*, *Csf3*, *Cxcl1*, *Ccl2*, *Gata3*, *Il5*, and *Il13*), there was a clear difference between the EC and HC *Shb* KO effects using *Cdh5-Cre^ERt2^* or *Vav1-Cre* tumor-bearing mice, indicating that these effects reflect EC cell autonomy of *Shb* gene deletion. The reduction of *Ifng* expression was seen with both Cre transgenes, suggesting that this effect is cell-autonomous to HC, whereas for the remaining changes, an uncertainty remains as to which cell type they reflect.

The gene expression data have been presented according to established roles as immune response genes (*Ifng* [23] and *Il10* [24]), MDSC recruiting and/or expanding cytokine/chemokine genes (*Csf3*, *Cxcl1*, *Ccl2* [13,25,26]), immune checkpoint genes (*Ctla4* and *Cd274* = Programmed Cell Death Ligand 1 or PD-L1), and Th2 genes (*Gata3*, *Il5* and *Il13*). Shb deficiency reduced *Ifng* and increased *Il10* gene expression (Figure 5), suggesting an immunosuppressive tumor microenvironment relative to the wild type situation, since IFNγ is known to enhance and IL-10 to suppress immune responses. *Csf3*, *Cxcl1*, and *Ccl2* were all increased in EC *Shb* KO mice, providing a plausible explanation for the increased mMDSC infiltration (Figure 5). EC *Shb* KO reduced the expression of immune checkpoint genes *Ctla4* and PD-L1 (*Cd274*) in tumor CD45+ cells, suggesting that the observed reduction in the expression of these genes plays a minor role in the observed increase in metastasis. Finally, genes coding for Th2 cytokines were unexpectedly reduced in tumors from mice with EC *Shb* KO, an effect discrepant of that recorded with the HC knockout. Gene expression changes that were not significantly altered are shown in Table 2. The data thus demonstrate that *Shb* in EC influences the recruitment of immune cells in a manner that has an impact on the immune response towards the tumor. The combined data suggest that EC *Shb* gene expression will influence the local microenvironment in such a manner that recruitment/expansion of mMDSC is suppressed, allowing an anti-tumoral immune response to prevail.

## 3. Discussion

The data suggest that the main cause of increased tumor metastasis in EC *Shb KO* mice is the development of an immunosuppressive environment rather than a leaky vasculature. MDSC appear to be the primary cause of this immunosuppressed environment since mMDSC were increased in both the tumors and at metastatic sites in lungs in the absence of *Shb* expression in EC. Tregs may also contribute to a pro-tumoral environment, although their role is less obvious since they were not increased at metastatic sites. The decreased expression of immune checkpoint genes and Th2 cytokine genes appear to play a minor role in the current setting. The reduction of Th2 gene expression was an effect clearly discrepant from that previously observed using a global *Shb* KO. In that study, we observed an increased Th2 response [27,28], which is compatible with the finding of increased *Il5* expression in CD45^+^ tumor cells in HC *Shb* KO mice. The discrepant EC *Shb* KO response thus reflects effects on tumor infiltration of cells expressing Th2 cytokines.

MDSC exert their pro-tumoral immunosuppressive effects by various means [13]. These include the promotion of Tregs, secretion of IL-10 and TGFbeta, release of reactive oxygen species, peroxynitrite and arginase 1, downregulation of CD62L, and expression of PD-L1 [13]. With such a plethora of responses, it is difficult to pinpoint the exact mechanisms of action in the current setting, but we do confirm increased Treg numbers and increased gene expression of the immunosuppressive cytokine IL-10 in CD45^+^ cells from tumors grown in mice with *Shb* deficient EC. An additional contribution to this microenvironment could be the decreased expression of the pro-inflammatory cytokine IFNγ. However, a similar effect on IFNγ expression was observed in HC *Shb* KO cells, suggesting that this *Shb*-dependent effect may be cell-autonomous to *Shb* deficiency in HC.

CCL2 (C-C motif chemokine ligand 2), CXCL1 (C-X-C motif chemokine ligand 1), and G-CSF (granulocyte colony stimulating factor = *Csf3* gene) are all cyto- or chemokines that promote MDSC [13,25,26,29,30,31], and their expression was increased in CD45^+^ cells from *Shb* KO EC, providing a likely explanation for the MDSC-enriched microenvironment. G-CSF is primarily related to nMDSC expansion but may also promote myeloid progenitor expansion from which mMDSC may develop due to other cues [29]. The identity of the cell(s) expressing these cytokines has not been established but some modified EC feature caused by *Shb* deficiency would explain altered HC tumor homeostasis resulting in the observed cytokine profile.

A limitation of this study is that partial *Shb-*gene deletion in HC using the endothelial *Cdh5-Cre^ERt2^* line may additionally, via an uncharacterized HC population, exert indirect effects that promote EC-dependent mMDSC recruitment/expansion. Moreover, the study presents gene expression data without the corresponding protein levels. In addition, the mechanism by which EC regulate the recruitment of mMDSC remains elusive. Finally, the mechanism by which mMDSC reduce tumor metastasis has not been established.

*Shb*-deficiency in EC alters VEGF-receptor 2 (VEGFR2) stimulation of focal adhesion kinase (FAK) activity with loss of FAK-activation, increased basal FAK activity, and focal adhesion redistribution as a consequence [32]. FAK is important for the activity of EC adherens junctions [33] and thus the properties of junctions and their ability to support HC transmigration will have consequences for tumor immunity in a *Shb-* and FAK-dependent manner. In addition, tumors grown in mice with *Shb*-deficient EC exhibit increased hypoxia due to a decreased vascular density [18]. Anti-angiogenic treatment commonly results in tumor hypoxia and recruitment of immunosuppressive leukocytes [34,35]. Whether the recruitment of immunosuppressive leukocytes is the consequence of hypoxia itself or due to altered EC properties that skew the transmigratory properties of leukocytes is unknown but the current data conform to previous data on anti-angiogenic treatment in this context and add an element of immunosuppressive immune cell infiltration that depends of EC function to the concept of tumor escape from anti-angiogenic treatment [36].

RNAseq of *Shb* deficient tumor EC did not reveal any apparent cytokine/chemokine changes but rather gene ontology changes pertaining to focal adhesions and junctions, both of which were considered relevant to the observed decrease in vascular leakage [18]. Adherens junctions have been shown to operate as gatekeepers for leukocyte transmigration [37] and thus the gene expression data would support *Shb*-dependent adherens-junctions-related changes as responsible for the selectively altered properties of leukocyte extravasation in tumors, supporting the recruitment and/or expansion of mMDSC. Conversely, HC cells expressing Th2 cytokines are selectively excluded from tumor extravasation in the current setting. Another possibility is that the vascular niche favors the survival of anti-tumoral cytotoxic T cells [38]. Thus, *Shb*-deficient EC would not be expected to support a microenvironment that promotes an active anti-tumoral immune response. In our previous publication, we noted that relatively fewer T cells were juxtaposed to vascular structures in *Shb*-deficient mice [19], supporting this concept.

The knowledge that the *Shb*-gene restricts the access of immunosuppressive cells to the tumor microenvironment allows potential for manipulation. Knowledge of the precise nature of relevant cells and the exact process regulating their extravasation is warranted in further studies.

## 4. Materials and Methods

### 4.1. Mice and Cells

*Shb^flox/flox^* mice [22] or wild type mice were bred with mice carrying the *Cdh5-Cre^ERt2^* [39] (EC *Shb* KO) or *Vav1-Cre* (HC *Shb* KO) transgenes [40]. E0771.lmb cells were used for tumor experiments [41]. All animal experiments were approved by the animal ethics board at the local Uppsala County court (case number 5.8.18-15140/2017).

### 4.2. In Vivo Experiments

Prior to tumor cell injections, *Cdh5-Cre^ERt2^* mice (wild type or *Shb^flox/flox^*) were tamoxifen injected as described [22]. *Vav1-cre* mice (wild type or *Shb^flox/flox^*) were tumor-cell-injected without tamoxifen administration. Since the *Cdh5-Cre^ERt2^* transgene also deletes *Shb* in some HC, we needed a control to distinguish between EC and HC cell-autonomous effects. Rather than completing whole body irradiation and bone marrow transplantation, we decided to use the *Vav1-Cre* transgene to obtain almost complete *Shb* gene deletion in all HC. One million E0771.lmb cells were orthotopically injected in the subcutaneous fat pad in the fourth mammary gland. Tumor progression was regularly monitored and, at Day 14, wounds began developing over the tumors, which motivated resection on that day. The resected tumor was digested with collagenase, and CD45^+^ cells were MACS^TM^-purified (Miltenyi Biotech, Bergisch Gladbach, Germany), as described [18,22]. The mice were maintained for an additional 3 weeks, after which they were sacrificed. Lung surface metastasis was monitored by visual inspection. Local (inguinal) lymph nodes were also resected and cells prepared for flow cytometry. To analyze metastatic lungs, E0771.lmb cells (0.8 million) were tail-vein-injected. Mice were sacrificed three weeks later, metastasis-dense regions of the lungs were excised and digested, and CD45^+^ cells were MACS^TM^-purified. Appendix A lists all immunological reagents used.

### 4.3. Tumor Morphology

The tumor vasculature was assessed by staining frozen tumor sections for CD31, desmin, and fibrinogen as described [18]. Tumors from 5 WT (*Cdh5-Cre^ERt2^*) and 4 KO (*Shb^flox/flox^/Cdh5-Cre^ERt2^*) mice were analyzed (3–5 sections each tumor) by measuring the length of desmin staining as a percentage of vessel circumference or by using ImageJ software (version 1.51m9, National Institutes of Health, Bethesda, MD, USA) for determining the staining intensity (CD31 and fibrinogen). Density was percent of section area and leakage was percent extravascular fibrinogen relative intravascular fibrinogen. Appendix A lists all immunological reagents used.

### 4.4. Flow Cytometry

CD45^+^ cells from tumors or parts of metastatic lungs with a high degree of metastatic seeding or cells from lymph nodes were stained for flow cytometry as described [18,21] using FlowJo software (version 10.8.0, Becton Dickinson, Franklin Lakes, NJ, USA) and a Fortessa LSR flow cytometer (Becton Dickinson). Gating strategies for Tregs and mMDSC are shown in Appendix A. Appendix A lists all immunological reagents used.

### 4.5. Gene Expression

RNA was isolated from CD45^+^ cells using the Qiagen RNeasy mini-kit (Qiagen, Hilden, Germany). SYBR qPCR was performed using the Roche SYBR-kit (Roche, Basel, Switzerland) on a Roche Light Cycler. Light Cycler software was used to calculate Ct values. All Ct values were normalized for beta-actin as the loading control. Appendix A lists primer sequences.

### 4.6. Statistics

Means ± SEM for the number of observations are given. Each observation corresponds to one tumor mouse.

## Figures and Tables

**Figure 1 ijms-22-11478-f001:**
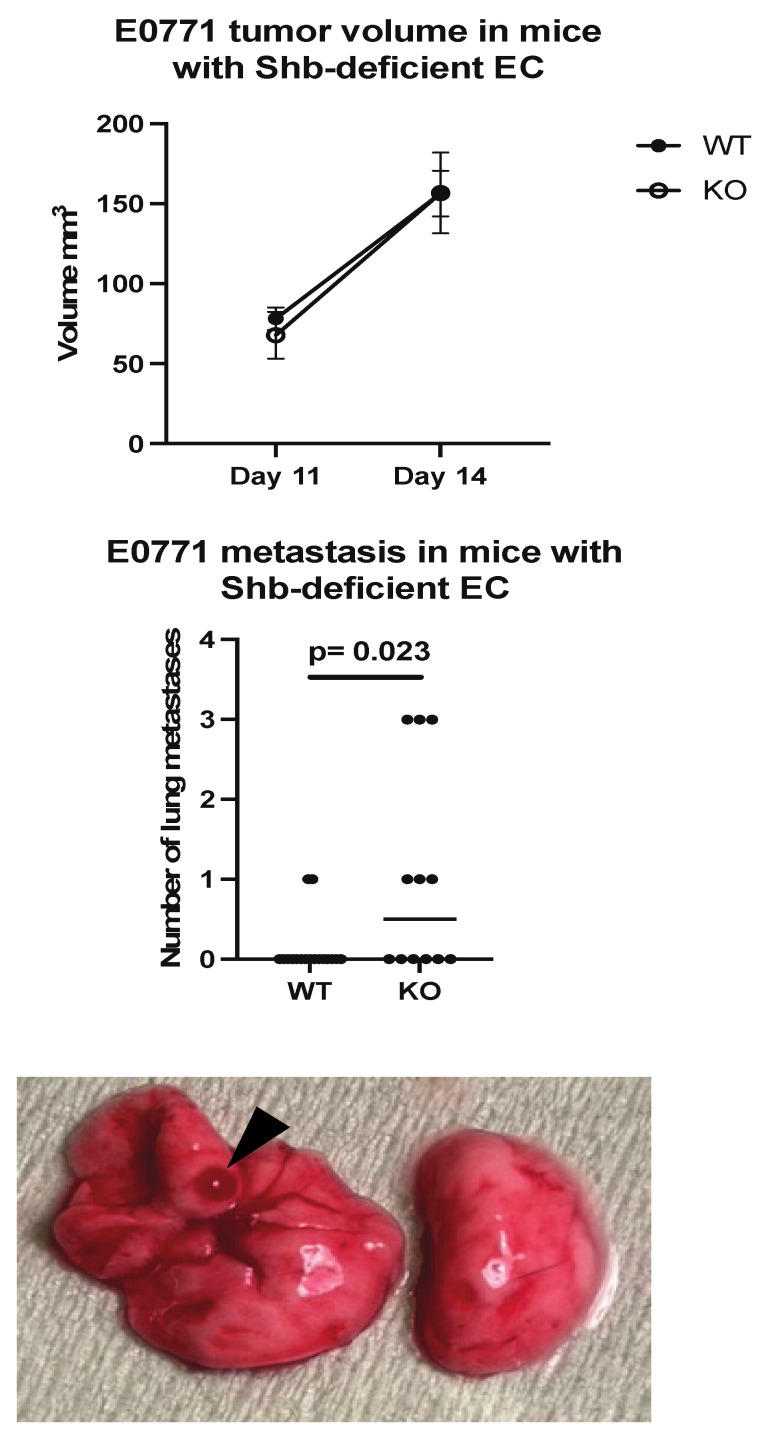
Tumor growth and metastasis in wild type (WT) and EC *Shb* knockout (KO) mice. WT *n* = 12 and KO *n* = 8 for tumor growth and 17 and 12 for metastasis, respectively. A Chi-square test was employed to compare WT and KO. A hemorrhagic metastasis is indicated by an arrow head.

**Figure 2 ijms-22-11478-f002:**
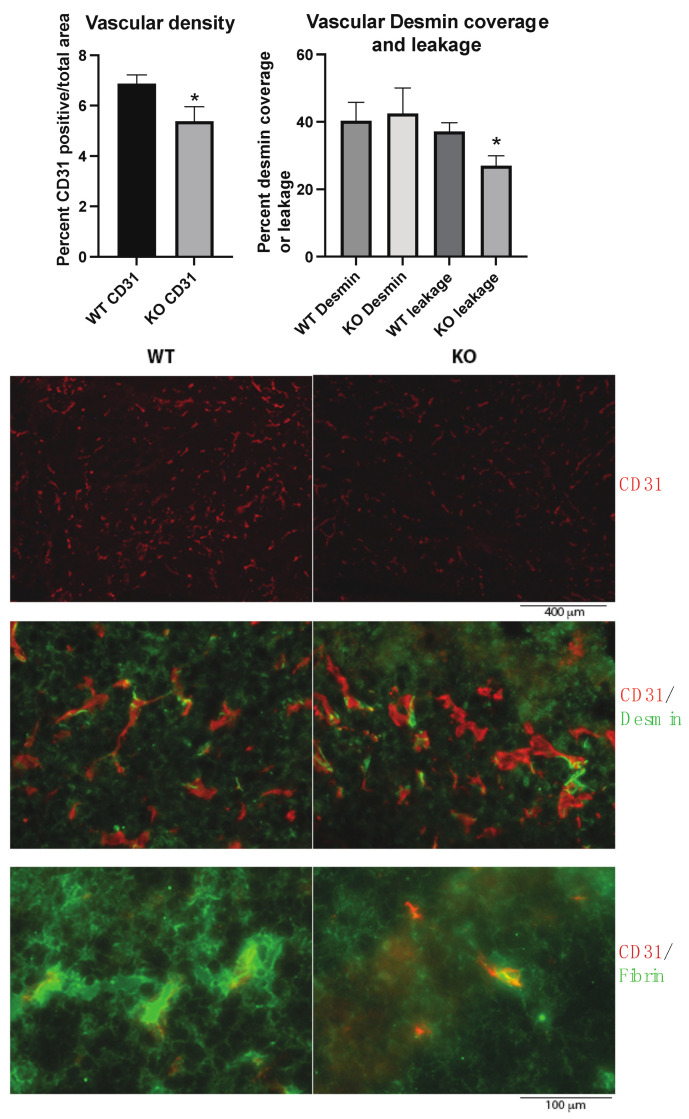
Tumor vasculature in wild type (WT) and EC *Shb* knockout (KO) mice. Vascular density as percentage of CD31-positive area, percent Desmin (pericyte) coverage of small vessel circumference, and leakage (extravascular Fibrin) were determined for 5 WT and 4 KO tumors/mice. In the images with higher magnification, areas with comparable vascular density were selected to allow representative assessment of Desmin coverage and leakage. Scale bar for CD31 staining alone is shown below images. CD31/Desmin and CD31/Fibrin stains have the same scale bar which is shown at the bottom. * indicates *p* < 0.05 by Student’s *t*-test.

**Figure 3 ijms-22-11478-f003:**
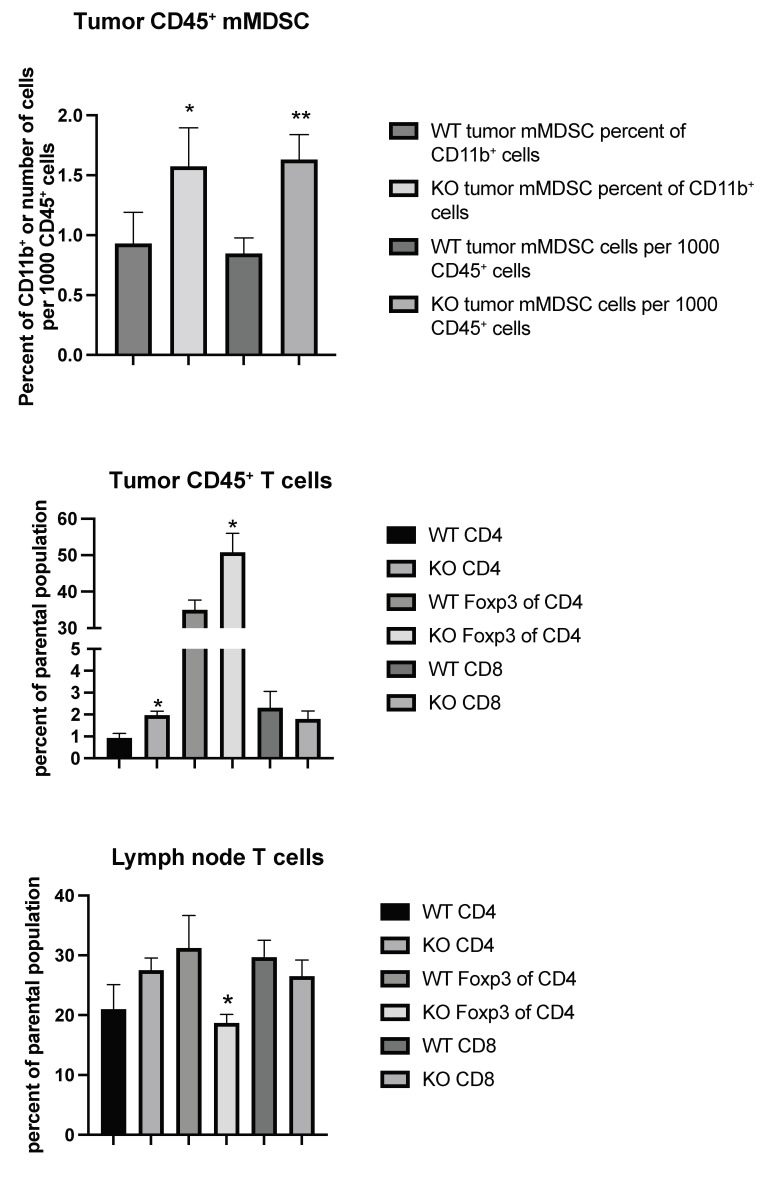
Tumor and inguinal lymph node immune cell profiling (mMDSC and T cells) using flow cytometry from wild type (WT) and EC *Shb* KO (KO) mice. Values are percent of parental population ± SEM. *n* = 6 for WT and *n* = 4 for KO. * and ** indicate *p* < 0.05 and 0.01 by Student’s *t*-test, respectively.

**Figure 4 ijms-22-11478-f004:**
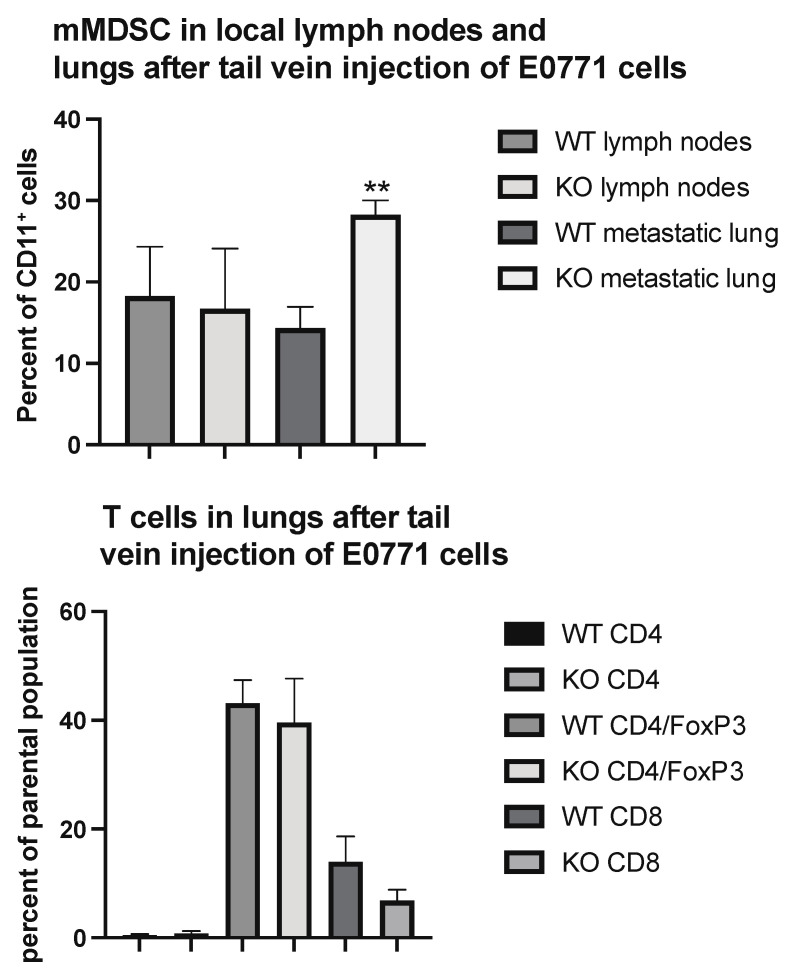
Immune cell profiling of lung areas with dense tumor cell colonization from wild type (WT) and EC *Shb* knockout (KO) mice using flow cytometry. Upper panel shows mMDSC and lower panel T cells. *n* = 6 mice for WT and 4 for KO. ** indicates *p* < 0.01 by Student’s *t*-test.

**Figure 5 ijms-22-11478-f005:**
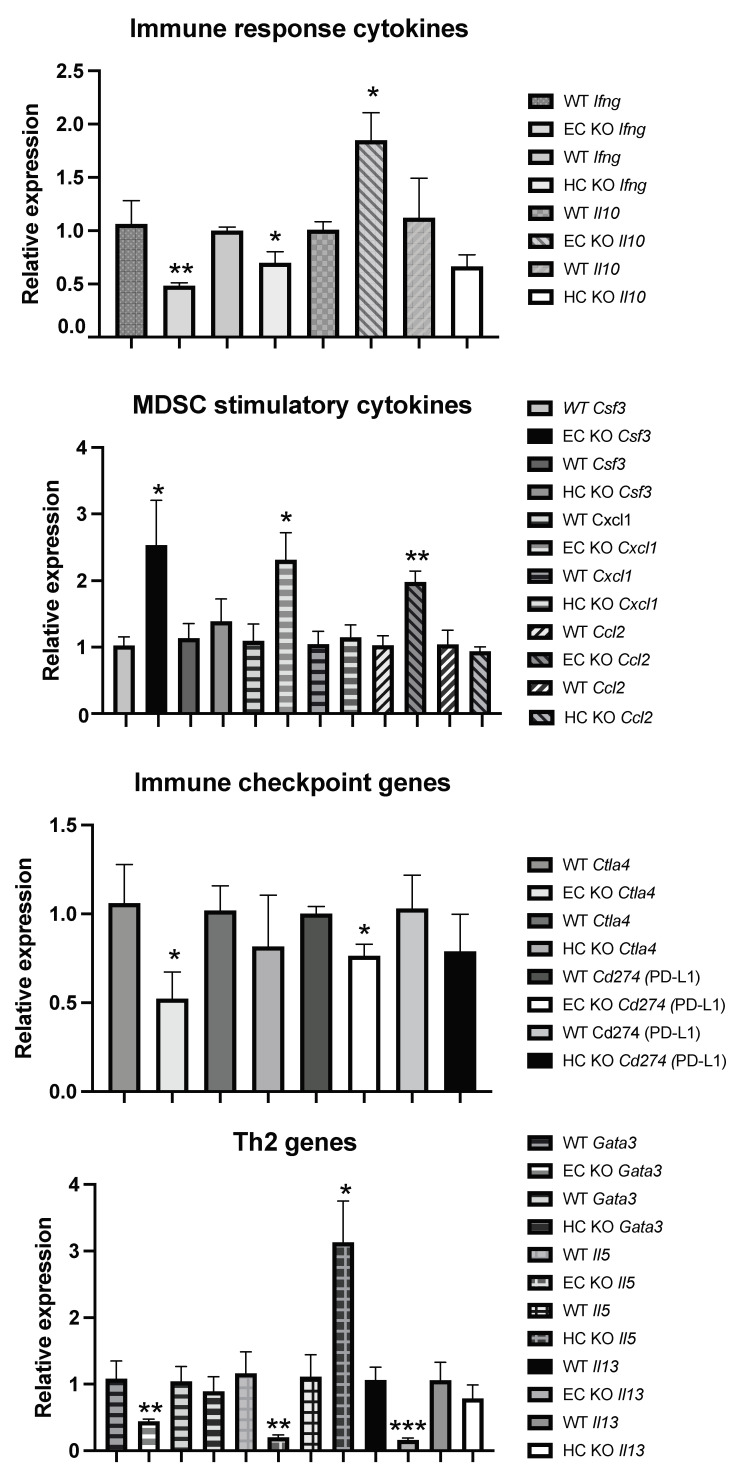
CD45^+^ cell gene expression of various genes related to immune cell function in tumors in mice from wild type (WT), EC *Shb* KO, and HC *Shb* KO mice. *n* = 4 for WT and *n* = 5 for EC KO. *n* = 3–9 for WT and *n* = 3–9 for HC KO. *, **, and *** indicate *p* < 0.05, *p* < 0.01, and *p* < 0.001, respectively, using a Student’s *t*-test.

**Table 1 ijms-22-11478-t001:** Additional immune cell populations determined by flow cytometry.

Population	WT (Percent of Parental ± SEM)	KO (Percent of Parental ± SEM)
Lymph node CD11b	4.9 ± 2.5 (10)	2.2 ± 0.5 (6)
Lymph node CD8 of CD11b	54 ± 8.7 (6)	50 ± 6.8 (4)
Lymph node CD11c	7.5 ± 1.9 (10)	4.6 ± 0.1 (6)
Lymph node CD8 of CD11c	38 ± 3.2 (6)	39 ± 2.0 (4)
Lymph node nMDSC	5.1 ± 1.2 (6)	4.7 ± 1.1 (4)
Lymph node mMDSC	25 ± 2.7 (6)	26 ± 3.3 (4)
Tumor CD11b	8.5 ± 0.5 (3)	9.0 ± 0.6 (3)
Tumor CD8 of CD11c	32 ± 2.2 (3)	28 ± 2.0 (3)
Tumor CD11c	1.5 ± 0.4 (6)	2.1 ± 0.3 (5)
Tumor CD8 of CD11c	16 ± 3.4 (6)	11 ± 1.2 (5)
Tumor nMDSC	0.4 ± 0.1 (6)	0.2 ± 0.1 (5)
Tumor CD11b^+^/Ly6G^+^	3.3 ± 0.6 (9)	5.1 ± 1.0 (6)

**Table 2 ijms-22-11478-t002:** Relative gene expression as Ct values in tumor CD45^+^ cells after subtraction of beta-actin values in wild type (WT) and EC *Shb* knockout (KO) cells. WT is based on four tumors/mice and KO five.

Gene	WT Ct ± SEM	KO Ct ± SEM	*p* Value
*Shb*	9.8 ± 0.63	11.4 ± 0.24	0.04
*Il4*	16.5 ± 0.22	16.3 ± 0.37	0.73
*Arg1*	5.6 ± 0.22	6.4 ± 0.15	0.01
*Il1b*	0.15 ± 0.32	0.0 ± 0.16	0.65
*Il6*	7.8 ± 0.19	8.3 ± 0.40	0.35
*Tbx21*	12.5 ± 0.80	13.2 ± 0.21	0.09
*Il17a*	14.3 ± 0.33	14.4 ± 0.44	1.00
*Il12a*	15.4 ± 0.35	14.7 ± 0.18	0.18
*Tnf*	11.1 ± 0.55	10.9 ± 0.27	0.07
*Nos2*	7.6 ± 0.28	7.5 ± 0.25	0.78
*Csf2*	14.3 ± 0.43	14.7 ± 0.33	0.49
*Csf1*	9.4 ± 0.59	10.0 ± 0.26	0.34
*Cd4*	12.1 ± 0.62	12.6 ± 0.09	0.41
*Cd8a*	12.9 ± 0.28	12.4 ± 0.27	0.27
*Foxp3*	8.4 ± 0.32	9.2 ± 0.24	0.07
*Gzmb*	13.1 ± 0.22	12.6 ± 0.23	0.16
*Pdcd1 (PD1)*	9.8 ± 0.10	10.3 ± 0.38	0.30
*Itgam (CD11b)*	7.2 ± 0.19	7.0 ± 0.41	0.78
*Adgre1 (F4/80)*	9.4 ± 0.19	8.8 ± 0.48	0.33
*Itgax (CD11c)*	7.9 ± 0.21	7.8 ± 0.32	0.76
*S100a9*	15.2 ± 0.6	15.0 ± 0.32	0.76
*Tgfb1*	8.3 ± 0.18	7.8 ± 0.21	0.21
*Tgfb2*	11.2 ± 0.16	10.9 ± 0.28	0.47
*Tgfb3*	12.8 ± 0.27	12.1 ± 0.27	0.11
*Cxcr4*	6.2 ± 0.24	6.2 ± 0.23	0.91
*Cxcl12*	13.6 ± 0.22	13.7 ± 0.43	0.84
*Cxcl9*	10.4 ± 0.22	9.8 ± 0.48	0.30
*Cxcl10*	4.6 ± 0.49	4.5 ± 0.22	0.80
*Cxcl11*	9.0 ± 0.50	8.9 ± 0.21	0.84
*Ccr7*	7.0 ± 0.23	6.4 ± 0.44	0.31
*Ccl3*	7.3 ± 0.16	7.0 ± 0.20	0.32
*Ccl4*	6.6 ± 0.25	6.2 ± 0.29	0.37
*Ccl20*	19.0 ± 1.00	18.5 ± 0.26	0.56
*Ccl21*	20.7 ± 0.79	21.4 ± 1.26	0.27
*Ccl22*	12.5 ± 0.16	12.1 ± 0.16	0.12

## Data Availability

Original data are available from the lead author upon request.

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
