# Peer review of "Mouse Breast Carcinoma Monocytic/Macrophagic Myeloid-Derived Suppressor Cell Infiltration as a Consequence of Endothelial Dysfunction in Shb-Deficient Endothelial Cells Increases Tumor Lung Metastasis"

_ijms, 2021, doi:10.3390/ijms222111478_

Round 1
Reviewer 1 Report
Authors present a work addressing: ”Endothelial Shb gene expression reduces mouse breast carcinoma monocytic/macrophagic myeloid-derived suppressor cell infiltration with consequences for tumor lung metastasis”. The general conclusion shown an intricate interplay between tumor EC and immune cells that pivots between pro-tumoral and an anti-tumoral properties, depending on relevant genetic and/or environmental factors operating in the microenvironment. The topic of the article is relevant in the field. I my opinion the manuscript is outstanding. Obviously, I endorse to publish this article once Authors correct my suggestions:
- The abstract should includes information related to methodology.
- I suggest to add p-value on the figure 1.
- The authors should provide limitations of the study.
Author Response
We thank the reviewers for their positive comments and have revised accordingly.
Reviewer 1: Many thanks for the appreciation and the constructive critique.
Information on methodology in the abstract has been added as follows:
Lung metastasis from orthotopic tumors, tumor vascular and immune cell characteristics and immune cell gene expression profiles were determined.
The p-value has been added (p=0.023)
The limitations of the study have been described as follows.
A limitation of this study is that partial Shb-gene deletion in HC using the endothelial Cdh5-CreERt2 line may via an uncharacterized HC population exert indirect effects that promote EC-dependent mMDSC recruitment/expansion. Furthermore, the study presents gene expression data without the corresponding protein levels. In addition, the mechanism by which EC regulate the recruitment of mMDSC remains elusive. Finally, the mechanism by which mMDSC reduce tumor metastasis has not been established.
Reviewer 2 Report
The issue is very interesting; however, I have some remarks:
The title does not clearly indicate the content.
Introduction section, lines 88-91: the aim must be better explained; Authors must not summarize their findings in this section.
Results section: the presentation is sometimes unclear, and this section contains also methods, comments, references.
In conclusion, I suggest Authors to re-organize their manuscript in order to make a more clear and readable presentation
Author Response
Reviewer 2: Many thanks for the appreciation and the constructive critique.
The title has been changed to “Mouse breast carcinoma monocytic/macrophagic myeloid-derived suppressor cell infiltration as a consequence of endothelial dysfunction in Shb-deficient endothelial cells increases tumor lung metastasis”
The aims (lines 93-100) have been revised to “Since the Shb-gene exerts regulatory roles in both EC and HC in a tumor context, and since breast cancer growth and metastasis are influenced by both these stromal cell types, we considered employing the EC-specific Shb-gene knockout model to obtain a better understanding of the role of these cell types and their interplay in promoting/suppressing tumor expansion. For this purpose, we assessed E0771.lmb lung metastasis in mice with endothelial Shb deficiency. We found that the immune system plays a major role in suppressing metastasis and that Shb deficiency in EC promotes the recruitment or expansion of an immune suppressed tumor microenvironment.”
Results have been revised and methods and comments have been minimized.
E0771.lmb tumor cells were orthotopically injected into wild type mice (Cdh5-CreERt2) or mice with the Shbgene conditionally deleted in EC (Shbflox/flox/Cdh5-CreERt2). This Cre transgene also deletes Shb in some HC [22]. There was no difference in tumor growth until day 14 (Fig. 1) at which ulcerations started to appear over some tumors prompting tumor resection. When the mice were sacrificed three weeks later, visual inspection of lungs revealed increased lung surface metastasis in mice with Shb absent in EC (Fig. 1). (103-114)
The tumor vasculature was investigated since vascular leakage is important for metastasis and the Shb gene has been shown to influence several EC and pericyte parameters. (121-123)
Considering the possibility that tumor immune responses may play a role in tumor metastasis, we.. (156-157)
To determine if these changes would reflect the situation at the lung metastatic seeding sites, we immune profiled lung CD45+ cells in parts of the lungs heavily colonized by tail vein injected tumor cells. As was the case in the primary tumor, the areas colonized by tumor cells exhibited an increased percentage mMDSC cells (Fig. 4) without affecting the presence of Tregs. (180-184)
and compared the results with those of HC deleted Shb using Vav1-Cre as a control to exclude off target effects of Cdh5-CreERt2 in HC. (190-192)
The gene expression data have been presented according to established roles as immune response genes (Ifng [23] and Il10 [24]), MDSC recruiting and/or expanding cytokine/chemokine genes (Csf3, Cxcl1, Ccl2[13, 25, 26]), immune checkpoint genes (Ctla4 and PD-L1=Cd274) and Th2 genes (Gata3, Il5 and Il13). (202-205)
Thus , the manuscript has been extensively organized and text inserted into discussion and methods:
The reduction of Th2 gene expression was an effect clearly discrepant from that previously observed using a global Shb KO. In that study we observed an increased Th2 response [27, 28], which is compatible with the finding of increased Il5 expression in CD45+ tumor cells in HC Shb KO mice. The discrepant EC Shb KO response thus reflects effects on tumor infiltration of cells expressing Th2 cytokines. (253-257)
Since the Cdh5-CreERt2 transgene also deletes Shb in some HC, we needed a control to distinguish between EC and HC cell-autonomous effects. Rather than completing whole body irradiation and bone marrow transplantation, we decided to use the Vav1-Cre transgene to obtain almost complete Shb gene deletion in all HC. (337-341)
Round 2
Reviewer 2 Report
Authors have only partially modified their paper; however, they have clarified the main points.
I suggest to revise carefully the text for some minor mistakes; I also suggest a little change:
Discussion
Page 7, Line 190 the phrase “The data indicate that the main…” should be replaced by “The data suggest that the main…”
Author Response
The suggestions have been carefully addressed. Page 7, line 156, "indicate" has been replaced by "suggest" as requested.
The manuscript has been proof read and minor mistakes corrected. On page 2, line 57, of has been replaced by in.
page 2, line 64, "role" has been deleted.
Page 2, line 74, "means" changed to " mechanisms"
Page 5, line 188-189, "affected" changed to "being affected".
Page 6, line 198, "off-target"
page 6, line 220, "the effect"
We thank the reviewer for observing these errors.